# Cortical stiffness of keratinocytes measured by lateral indentation with optical tweezers

**Špela Zemljič Jokhadar**[1], **Biljana Stojković**[1], **Marko Vidak**[2], **Tjaša Sorčan**[2], **Mirjana Liovic**[2], **Marcos Gouveia**[3], **Rui D. M. Travasso**[3], **Jure Derganc**[1]*

1 Institute for Biophysics, Faculty of Medicine, University of Ljubljana, Ljubljana, Slovenia, 2 Medical Center for Molecular Biology, Institute for Biochemistry, Faculty of Medicine, University of Ljubljana, Ljubljana, Slovenia, 3 Department of Physics, Centro de Física da Universidade de Coimbra (CFisUC), University of Coimbra, Coimbra, Portugal

* jure.derganc@mf.uni-lj.si

## Abstract

Keratin intermediate filaments are the principal structural element of epithelial cells. Their importance in providing bulk cellular stiffness is well recognized, but their role in the mechanics of cell cortex is less understood. In this study, we therefore compared the cortical stiffness of three keratinocyte lines: primary wild type cells (NHEK2), immortalized wild type cells (NEB1) and immortalized mutant cells (KEB7). The cortical stiffness was measured by lateral indentation of cells with AOD-steered optical tweezers without employing any moving mechanical elements. The method was validated on fixed cells and Cytochalasin-D treated cells to ensure that the observed variations in stiffness within a single cell line were not a consequence of low measurement precision. The measurements of the cortical stiffness showed that primary wild type cells were significantly stiffer than immortalized wild type cells, which was also detected in previous studies of bulk elasticity. In addition, a small difference between the mutant and the wild type cells was detected, showing that mutation of keratin impacts also the cell cortex. Thus, our results indicate that the role of keratins in cortical stiffness is not negligible and call for further investigation of the mechanical interactions between keratins and elements of the cell cortex.

## Introduction

Mechanical stiffness of animal cells is provided by the cytoskeleton composed of three main structural elements: actin filaments (AF), microtubules (MT) and intermediate filaments (IF) [1]. MT and IF predominantly reside in the cell interior, whereas AF compose the submembrane acto-myosin cortex that provides the cortical tension on the cell surface [2]. In a functional cytoskeleton, all three filament systems are not only intertwined but also have a large number of associated proteins that interact or bind to them (e.g., motor proteins, focal adhesion proteins, desmosome and hemidesmosome proteins, tight junction proteins, etc.) [3, 4]. In such a complex structure, a disruption of one type of filaments can cause rearrangement of the others [5], and *in vivo* this can also lead to diseases.

In epithelial cells, the incidence of IF can be much higher than that of AF and MT, especially in the perinuclear region [6]. The important role of IF can be witnessed in karatinocytes

**Funding:** This work resulted from a collaboration initiated by the COST Action CA15214 (EuroCellNet). MG and RDMT thank the support of FEDER funds through the Operational Program Competitiveness Factors - COMPETE and by national funds by FCT - Foundation for Science and Technology under the strategic project UID/FIS/04564/2016 and under PTDC/BIA-CEL/31743/2017. MG thanks support by UE/FEDER funds through the program COMPETE 2020, under the project CENTRO-01-0145-FEDER-000014 (MATIS). RDMT acknowledges FCT's support through the FCT Researcher Program. The work was also supported by Slovenian Research Agency Program Grants P1-0055 (JD, BS and SZJ)and P1-0390 (ML). This work was supported by the Republic of Slovenia Ministry of Education, Science and Sport and the EraCoSysMed (JTC-2 2017) "4D-HEALING" grant, and the Celsa Alliance grant to ML (prof Radovan Komel). None of the funders plaid any role in the study design, data collection and analysis, decision to publish, or preparation of the manuscript.

**Competing interests:** The authors have declared that no competing interests exist.

—epithelial cells that form the skin's outer shell—where a mutation of a single IF protein (either keratin 5 or keratin 14) causes a genetic disease epidermolysis bullosa simplex, which manifests in the inability of skin keratinocytes to resist physical stress and induces severe skin lesions that may even lead to death. A typical characteristic of those keratin mutants is the presence of highly dynamic keratin particles in the cell periphery [7–10].

It has been shown that vimentin IF contribute significantly to intracellular mechanics but little to the stiffness of the sub-membrane cortex [11]. Atomic force microscopy (AFM) on keratinocytes have demonstrated that a mutation or knock-out of specific keratin IF also significantly impacts cell stiffness against deformations that penetrate deep into the cell [12, 13], but the role of keratin in cell mechanics under small deformations has not been investigated. As keratins interact biochemically with actin filaments and contribute to focal adhesions at the cell surface [14], one can hypothesize that the distinctive accumulation of keratin particles at the cell periphery in mutant cells might be related to altered mechanical properties at the level of cell cortex and plasma membrane.

To examine, if keratin IF contribute to the mechanical properties of the cell cortex, we set-up a method based on optical tweezers (OT) for indenting the cells gently from the lateral side with deformations in the range of a few 100 nm. The cell indentation was carried out by steering an optically trapped microbead with acousto-optic deflectors (AOD) without employing any moving mechanical part. The measurements were performed on three cell lines: NEB1 (an immortalized wild type cell line), KEB7 (an immortalized K14 R125P mutant cell line) and NHEK2 (primary wild type cells). It was found that the cortical stiffness of primary cells was significantly higher than one of immortalized wild type cells, which is in line with previous studies of bulk stiffness. Interestingly, a small difference between the mutant and the wild type immortalized cells was also detected.

## Materials and methods

### Cell lines and sample preparation

The study focused two previously extensively characterized immortalized keratinocyte cell lines [15, 16], which were derived from skin punch biopsies of a healthy individual (NEB1 cell line) and an epidermolysis bullosa simplex patient with a severe phenotype (KEB7 cell line, expressing the K14 R125P mutation). Both cell lines were first described in Morely et all [7] and were a kind gift from prof. E.B. Lane. The third cell line studied were primary cells (NHEK2) obtained from a control/healthy anonymous individual and was not obtained exactly for this study (Ethical approval NHS, National Ethics Research Committee London (11/LO/0295, 29th July 2011).

All cells were grown in serum-free EpiLife medium supplemented with EpiLife defined growth supplement (1%) and gentamicin/amphotericin (Cascade Biologics, Thermo Fischer Scientific, Waltham, MA, USA), at 37˚C and 5% $CO_2$. EpiLife was prepared with calcium chloride. Mammary gland/breast epithelial cells derived from the metastatic site (MDA-MB-231) were obtained from ATCC (USA). MDA-MB-231 cells were grown in RPMI 1640 medium (Genaxxon bioscience, Germany) supplemented with 4.5 g/l of glucose, 2 mM 127 L-glutamine, 1 mM pyruvate and 10% fetal bovine serum (FBS; Sigma-Aldrich). They were maintained at 37˚C in a humidified atmosphere with 5% (v/v) CO2. To validate the method, the experiments were performed as well on MDA-MB-231 cells first treated with 10 µM Cytochalasin D (Sigma-Aldrich) for 15 min at 37˚C, and on NEB-1 cells fixed with a 4% paraformaldehyde for 20 min at 37˚C.

24 h before measurements all the cells for the experiments were seeded in custom-made sample chambers. They were assembled from an acetone cleaned, uncoated, glass coverslip to which a PDMS insert was attached using plasma.

## Immunofluorescence

To assess the distribution of actin filaments, microtubules and keratin 14 the cells were rinsed three times with PBST (PBS with added Tween 20), which was used for all subsequent washings. The fixation was done in warm 4% paraformaldehyde in PBS. After the fixation, 0.1% Triton-X-100 was used for the permeabilization of the cell membrane and then the non-specific binding sites were blocked in PBST containing 1% BSA. Antibodies against anti-tubulin and anti-cytokeratin 14 (ab52866 and ab7800, Abcam), were added to the blocking solution and left for 2h at room temperature. After washing, cells were incubated with secondary antibodies marked either with Alexafluor 488 or Alexafluor 568 (Thermo Fisher Scientific). For aktin labeling the cells were also fixed in warm 4% paraformaldehyde in PBS and after washing, phalloidin Alexafluor 488 (A12379, Thermo Fisher Scientific) in PBS was added to the samples for 30 min at room temperature.

The selected antibody against cytokeratin 14 targets the non-helical tail domain, which does not coincide with the mutation in KEB7 cells, so it should bind equally well to filaments in mutated and WT cells.

## Protein extraction and Western blotting

All of the growth medium was removed from the culturing flasks and then ice cold RIPA buffer (Sigma-Aldrich) complemented with protease inhibitors was added to the cells. After a minute the cells were carefully scraped from the bottom. The cell suspension was collected and centrifuged for 10 min (12,500 rpm) at 4˚C. The protein concentration was determined by Pierce™ BCA Protein Assay Kit (Thermo Fischer Scientific, Waltham, MA, USA). Equal amounts of protein (20 μg) were separated on a polyacrylamide gel (NuPAGE Novex Bis -Tris gel 4–12%, Invitrogen, Thermo Fisher Scientific) and separated using electrophoresis. Subsequently, proteins were transferred to PVDF membrane. Membranes were blocked in 5% (w/v) skimmed milk in TBS-T (20 mM Tris, 150 mM NaCl, 0.02% Tween-20, pH 7.5), which was followed by overnight incubation in primary antibodies against keratin 14 (ab7800, Abcam) at room temperature and for the control anti GAPDH (CY-5174T, Cell Signaling) was used. After washing, membranes were incubated with the appropriate secondary antibodies for 1 hour at room temperature. The proteins were detected with a LAS-400 analyzer. The reported results are the mean over three independent experiments.

## Optical tweezers

Cell deformation experiments were performed on an Eclipse Ti inverted microscope (Nikon, Tokyo, Japan) equipped with laser tweezers (Tweez 250si, Aresis, Ljubljana, Slovenia). The optical tweezers setup used an infrared laser beam ($\lambda$ = 1064 nm), which was set on a constant optimal power so that there was no evidence of cell damage or bleb formation. The laser beam was focused through a water immersion objective (60x, NA 1.00, Nikon) into a sample chamber, which was maintained at a constant temperature (37˚C). The laser trap position and velocity were computer-controlled with acousto-optic deflectors (AOD) with 100 kHz rate. The laser power was calibrated according to the manufacturer's protocol to be uniform across the central part of the field of view, where the force measurements were performed. The experiments were recorded at approximately 100 frames/s with a CMOS camera (PLB-741, PixeLink, Montreal, Canada) under conventional brightfield illumination and the bead position was

tracked by TweezPal software [17]. Before each set of experiments, the stiffness of the laser trap ($\alpha$) was determined by the equipartition theorem from at least 30000 images of Brownian motion of a bead in a stationary trap [18]. In a typical experiment, the trap spring constant was $\alpha$ = 140 pN/μm.

## Indentation experiments

Streptavidin-coated silica microbeads with a radius of a 5.06 μm (CS01N, Bangs Labs, Fishers, IN) were used for indentation experiments. The beads were functionalized by an anti-integrin beta 1 antibody (ab28100, Abcam) to increase the adhesion of the bead to the cell membrane. A custom-made sample chamber containing adherent keratinocytes was mounted on the microscope and the beads were added to the sample. A bead was trapped by the optical twee-zers approximately 2 μm above the bottom of the chamber and a single, non-confluent cell was positioned at a predefined point near the bead by moving the microscope stage. The bead was facing the cell side where the cell membrane was most vertical, i.e., the side without extensive lamellipodia. The bead was then pushed laterally into the cell by moving the position of the optical trap at a constant velocity, and then retracted at the predefined point with the same constant velocity (Fig 1A and 1B, S1 Movie). The force-deformation curve was determined from the known positions of the trap and the bead, where the force on the bead was calculated from the displacement of the bead from the trap center ($\Delta x$) as $F = \alpha \Delta x$ (Fig 1C). The stiffness of the cell was quantified as the slope of the force-deformation curve [19] in the linear regime at the indentation of 200 nm. If the bead adhered to the membrane, the maximal force during the retraction, $F_{max}$, was measured. In a subset of adhesion incidents, a thin membrane tether was pulled from the cell membrane during retraction. To ensure the integrity of cells during the experiment, the measurements on each sample were performed within approximately 30 min, and the sample was then discarded.

To test for a possible viscoelastic response, i.e., the dependence of the apparent cell stiffness on the rate of deformation, the measurements were performed at bead velocities of 0.1 and 1 μm/s. These velocities corresponded to the slowest and the fastest velocity attainable by the

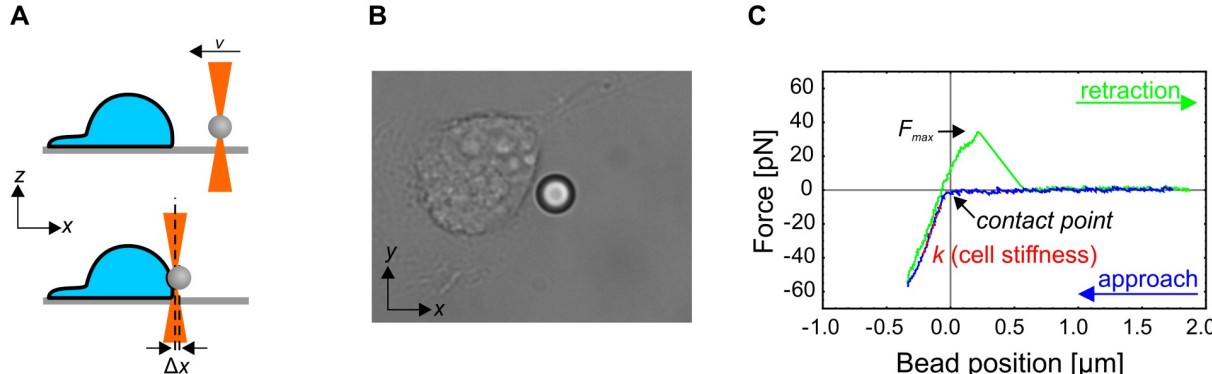

**Fig 1. Experimental protocol.** (A) A schematic representation of the experiment (side view). A silica microbead is trapped by optical tweezers and pushed into the cell at a constant speed by moving the optical trap with acousto-optic deflectors (top). The force exerted on the cell by the bead is determined from the displacement of the bead from the center of the optical trap ($\Delta x$; bottom). The bead is then retracted from the cell at the same constant speed. (B) A brightfield microscope image of the cell and the bead during the indentation experiment. The bead diameter is 5.06 μm. The whole experimental sequence is shown in S1 Movie. (C) A typical force-deformation curve. The cell stiffness ($k$) is determined as the slope of the linear part of the curve during the pushing phase. The adhesion of the bead onto the membrane is revealed as a maximal positive force ($F_{max}$) during retraction. In some cases, the force during the retraction never decreased to zero, which indicated that a membrane tether was extracted from the cell.

experimental set up (the high velocity is limited by the camera recording speed and the low velocity by the typical duration of the experiment for which the optical trap does not attract debris present in the sample). For trap velocities of 1 μm/s, the measurement was repeated twice at the same position.

### Statistical analysis

The statistical analysis were performed by Mathematica (Wolfram Research). Because the cell stiffness data did not exhibit a normal distribution, the Mann-Whitney-Wilcoxon test was used to calculate the p-values for comparison of the median cell stiffness between two data sets, and $p < 0.05$ was considered a significant difference (note that, if the student t-test was used to calculate the p-values, they were $< 0.05$ for almost all comparisons). The differences in variances were tested by the Variance Equivalence Test. The nominal data for adhesion events was tested by the Fisher's exact test.

## Results

### Distribution of cytoskeletal elements

The mechanical properties of cells are closely related to the distribution of the cytoskeletal elements. We therefore applied confocal microscopy to assess the distribution of the three main cytoskeletal elements (actin, tubulin, keratin 14) in all three cell types (NEB1, KEB7 and NHEK2). Althoug there is alvays some variability among cells even within one cell line, the most frequent distributions are presented in Fig 2. Within the resolution of the microscope, no unexpected features were detected. The distribution of actin filaments and tubulins appeared normal and did not differ appreciably among the cell lines (Fig 2A–2F). On the other hand, the distribution of keratin 14 in most keratin mutant KEB7 cells deviated significantly compared to the wild type cells NEB1 and NHEK2 (Fig 2G–2I). While keratin filaments in wild

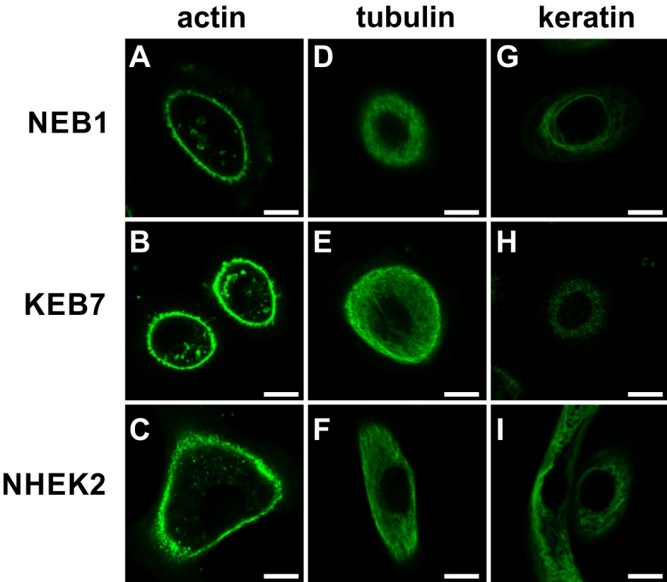

**Fig 2. The distribution of cytoskeletal elements.** Actin filaments (A-C), microtubules (D-F) and keratin 14 (G-I) in NEB1 (A, D, G), KEB7 (B, E, H) and NHEK2 cells (C, F, I) at the cell height where the microbead presumably indents the cell (approximately 5 μm above the bottom) are imaged. Scale bars represent 10 μm.

type cells exhibited the expected curly pattern, the filaments in most of KEB7 cells were missing, and the keratin signal was weak and dispersed throughout the cytoplasm. In very few KEB7 cells, some filaments could still be detected in the perinuclear region, which indicated that there was a slight variation in the KEB7 phenotype (S2 Fig).

Next, we performed western blot of K14 in all three cell lines (S4 Fig). Quantification of the western blot showed that mutant KEB7 cells exhibited approximately 5 times less keratin than NEB1 cells, which in turn had approximately 20% less keratin than NHEK2 cells. On the other hand the amount of the housekeeping protein GAPDH was equal in all three cell lines. Taken together, these results confirmed that the main structural difference between the three cell lines was in the severely impaired keratin filaments in the mutant cell line KEB7.

## Experimental protocol validation

Probing cell stiffness with optical tweezers allows repeated measurements on the same spot of the same cell and under the same experimental conditions. During the setting up of the experimental protocol on live keratinocytes, we not only observed considerable variations between individual cells but also between two consecutive measurements on the same spot of the same cell (Fig 3A). To verify that the observed variations were a consequence of the continuous remodeling of the cytoskeleton rather than of inaccurate measurements, we validated the experimental protocol also on fixed NEB1 cells and found good repeatability of the force-deformation curves on the same spot of one cell (Fig 3B). The variations in stiffness between two consecutive measurements on the same cell were significantly higher for live than for fixed cells (p<0.005, Fig 3A), indicating that the experimental approach is susceptible to continuous remodeling of live cells, which was evident during indentation experiments (S1 Movie). In the case of KEB7, NHEK2, and fixed NEB1 cells, the median difference between two consecutive measurements was less than 6%, while in the case of NEB1 the median difference was 38%, meaning that cell stiffness of NEB1 cells decreased from the first to the second measurement.

To further verify our experimental protocol against a positive control on living cells, we also measured the stiffness of Cytochalasin D treated cells (Cytochalasin D disrupts actin filaments making the cells softer). However, as keratinocytes have been shown to exhibit only a

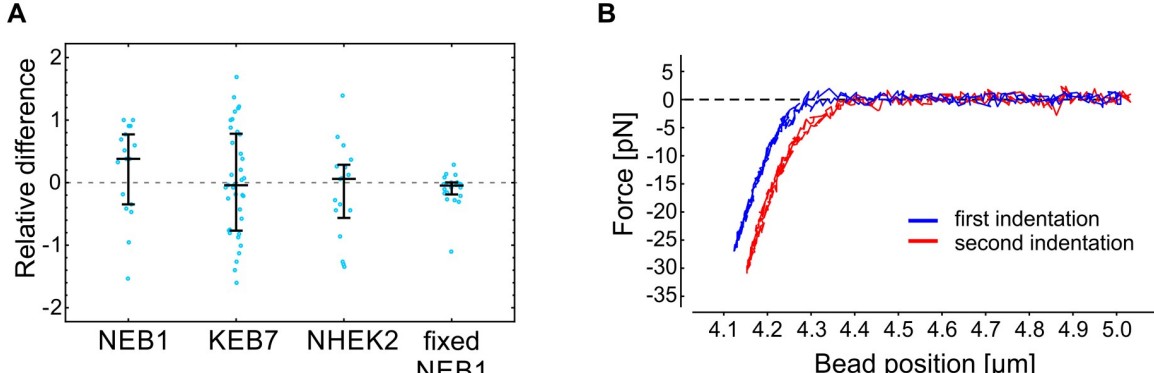

**Fig 3. Repeatability of the cell stiffness measurements.** (A) The relative difference between two consecutive measurements of the cell stiffness on the same spot of individual cells ($\frac{k_1 - k_2}{(k_2 + k_1)/2}$, where $k_1$ and $k_2$ are the stiffness recorded in the first and the second measurement, respectively; see Fig 1 for the definition of $k$). Measurements on living cells show a substantial variation, most likely due to active remodeling of living cells. On the other hand, the stiffness measurements performed on paraformaldehyde fixed cells indicate exceptional experimental reproducibility. The deformation rate in these experiments was set to 1 μm/s. The median value and the quartiles are denoted for each cell type. (B) A typical example of two force-deformation curves obtained on the same spot of a fixed cell. For clarity, the curves are shifted horizontally with respect to each other.

weak response to Cytochalasin D at small deformations [13], we measured the stiffness of MDA-MB-231 breast cancer cells before and after Cytochalasin D treatment. We chose this cell line because it is one of the standard lines in our lab and because cancer cells are known to respond to Cytochalasin D treatment [20]. The obtained results (S1 Fig) are in accordance with previous reports, i.e., the observed softening of MDA-MB-231 cells due to Cytochalasin D treatment was approximately 30% [13, 20].

We compared the stiffness of an immortalized wild type cell line (NEB1) with an immortalized keratin mutant cell line (KEB7), as well as with the stiffness of wild type primary cells (NHEK2) (Fig 3 and S1 Table). To test for a possible viscoelastic behavior, we performed the indentation experiments at deformation rates of 0.1 μm/s and 1 μm/s. Only measurements that exhibited a distinctive force-deformation curve (Fig 1C) were taken into account. We found that the stiffness of severe keratin mutant KEB7 cells was higher than that of immortalized wild type cells NEB1 when the lower bead velocity was used (p<0.005 at 0.1 μm/s) but not significantly different at the higher bead velocity (p = 0.21 at 1 μm/s). The stiffness of primary wild type keratinocytes NHEK2 was significantly higher than that of the immortalized wild type cell line NEB1 at both deformation rates (p<0.005), which is in line with the previously published data on AFM experiments [21]. As expected, the stiffness of fixed cells is significantly higher than the one of any of the live cells (p<0.005).

In general, cells do not respond to mechanical stress as pure elastic bodies but exhibit a passive flow or an active remodeling of the cytoskeleton with timescales from several tenths of a second to several minutes [2]. With the deformation rate of 0.1 μm/s, the duration of the cell indentation is in the order of magnitude of seconds, which may cause cortex remodeling and apparent cell softening, as typically observed in experiments on cellular mechanics [20, 22]. However, the measured effect of the deformation rate on cell stiffness was small when compared to variations within a given rate (Fig 4). The only statistically significant cell softening at a slow deformation rate was observed for NEB1 cells (78 pN/μm at 0.1 μm/s vs. 105 pN/μm at 1 μm/s).

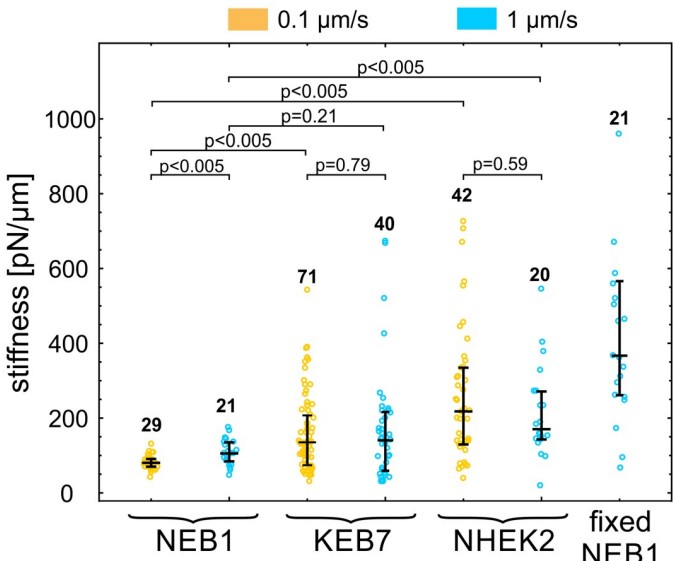

**Fig 4. Cell stiffness of three keratinocyte types.** Cell stiffness of NEB1, KEB7 and NHEK2 cells was measured at two deformation rates 0.1 μm/s (orange) and 1 μm/s (blue). Stiffness of fixed NEB1 cells was measured at 1 μm/s. The median value and quartiles are indicated for each cell type, the numbers of measured cells are above the data points, and the p-values of relevant pairs are specified at the top.

### Bead adhesion and membrane tether formation

Keratins have been shown to participate in focal adhesions at the surface of the cells [23, 24]. To verify, if the mutations affect also the adhesion between the plasma membrane and the bead, coated with anti-integrin beta 1 antibodies, we quantified the adhesion events in the indentation experiments (this adhesion can be detected as a positive force during bead retraction, Fig 1C). The incidence of adhesion in indentation experiments on different cell types is presented in Fig 5A. As expected, in all cases, a stronger adhesion was detected at the lower deformation rate due to a longer time of contact between the bead and the membrane. Wild type cells (NEB1) exhibited a larger proportion of adhesion events than the mutant cells (KEB7) (p = 0.002 at 0.1 μm/s and p = 0.17 at 1 μm/s). In all cell lines, the tether formation occurred in approximately 50% of adhesion events. In these experiments the relative magnitudes of the maximal force $F_{max}$ (Fig 5B) during bond rupture correlated with the adhesion strength and were larger for lower deformation rates. In the case of fixed cells, some adhesion between the bead and the cell also occurred, indicating the presence of non-specific adhesion, but no membrane tethers were formed (data not shown).

## Discussion

Data for vimentin IFs indicated that its disruption influences primarily the stiffness of the cell interior but not the stiffness of the cell cortex [11], and that the measured cell stiffness depends on the indentation depth [25]. AFM studies on keratinocytes also showed that keratin disruption has a significant impact on cell stiffness under large deformations [12, 13, 21], but the role of keratin at small deformations that penetrate only a few hundreds of nanometers into the cell remained unexplored. The goal of our study was to compare the cortical stiffness of three keratinocyte cell lines, and in particular if the stiffness is influenced by the incidence of keratin particles in the periphery of mutant cells [7–10].

In established methods for assessing cell mechanics, the cells are typically indented from the top by moving an AFM cantilever or a piezo-stage with fixed OT. In the approach presented in this paper, the cells were indented from the lateral side by AOD-steered optical tweezers without employing moving mechanical components (Fig 1), which can be an advantage in

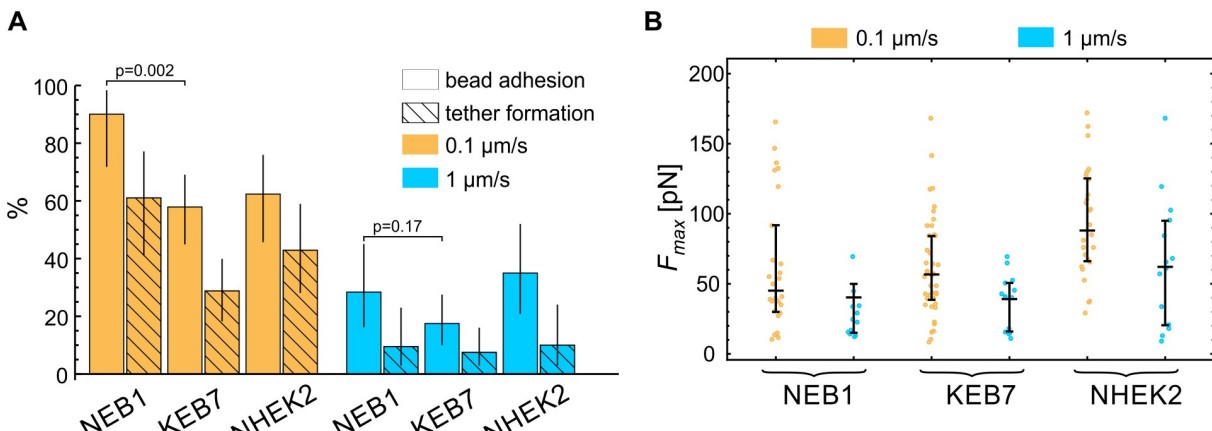

**Fig 5. Bead-membrane adhesion.** (A) The percentage of experiments with manifested bead adhesion on the cell membrane and subsequent membrane tether formation. A stronger adhesion was detected at the lower deformation rate due to a longer time of contact between the bead and the membrane. The vertical lines represent 95% confidence intervals. (B) The maximal force $F_{max}$ during bead retraction for experiments with manifested bead adhesion. $F_{max}$ during bond rupture correlated with the adhesion strength and was larger for lower deformation rates. The median value and the quartiles are denoted for each measurement.

many closed experimental systems, e.g., in microfluidic devices. The use of lateral indentation and a relatively large 5 μm microbead as an indenter allowed for straightforward microbead tracking and accurate measurement of the force-deformation curve. On the downside, the indentation could only be performed on the side of the cell without lamellipodia, where the cell geometry was relatively upright. Therefore, this approach is not suitable for analyzing the leading edge of migrating keratinocytes, which can markedly different from the trailing edge in terms of cell stiffness [26] and membrane tension [27]. Nonetheless, it should be noted that the cells used in our study were relatively immobile and only rarely exhibited the typical morphology of migrating cells, i.e., the one resembling a fried egg cut into a crescent shape with distinct leading/trailing edges (time-lapse recording of typical cells are presented the S2–S4 Movies).

In AFM experiments, the cell stiffness is often modeled by a Hertz model [28], which is sensitive with respect to inaccuracies in resolving the point of contact between the indenter and the cell. However, in our experiments the point of contact was hard to determine accurately, possibly due to the presence of a glycocalyx, which surrounds the cells and obscures their boundary. We therefore quantified the cell stiffness as the slope of the force-deformation curve in the linear part at a 200 nm indentation (Fig 1C). The value of this stiffness cannot be directly compared to the value of the Young modulus measured with AFM, but it is nevertheless a robust way to compare the cortical stiffness of different cell lines and to compare relative effects of cell treatments [2].

Large variations in the measured stiffness among cells within the same cell line were observed, possibly resulting from physiological and geometrical differences, which are always present in a population of living cells. In addition, there were variations between two consecutive measurements on the same spot of one cell (Fig 3), probably due to active remodeling of cells, which was apparent during experiments (S1 Movie). To validate the protocol, we therefore measured the stiffness of paraformaldehyde (PFA) fixed cells and found that it was indeed significantly higher than the one of living cells as well as repeatable (Fig 3). Cell stiffening upon PFA fixation had been shown also in AFM measurements [29]. The protocol was further validated by measuring the stiffness of cells treated with Cytochalasin D, and the results (S1 Fig) were in line with published data [13, 20]. In general, the variability among mutant KEB7 and primary NHEK2 cells was larger than the one among wild type NEB1 cells. Consequently, the latter was also the only cell type exhibiting a statistically significant dependence of the cell stiffness on the deformation rate.

The results of indentation experiments (Fig 4) revealed that the cortex in mutant keratinocytes (KEB7) was slightly stiffer than in the wild type cells (NEB1), but this stiffening was not statistically significant at the higher deformation rate and much less pronounced than the softening of the cell interior found in the studies by AFM [12, 13, 21]. Thus, these results strengthen the view that the main mechanical role of IF is in providing the bulk cellular stiffness. Still, IF could influence the cortical mechanics through interactions with other cytoskeletal elements, notably with the acto-myosin cortex [30]. In our study, the keratin signal was distributed in the cell interior underneath the actin cortex and there was no notable colocalization of keratin and actin (S3 Fig). Still, actin filaments are known to be involved in keratin turnover, e.g., it has been shown that keratin particles originate close to the plasma membrane, at the distal tips of actin stress fibers, and then they translocate continuously along the stress fibers toward the cell center, where they integrate into keratin filament network [23, 31]. Upon Cytochalasin-D treatment, which disrupts the actin network, the majority of keratin particles at the cells periphery either disappear or stop almost completely [32]. Also, it has been shown that keratin is closely related to structures at the cell surface, e.g., to focal adhesions [14, 23]. Indeed, our experiments revealed that keratin mutations caused a reduction in adhesion

between the membrane and the bead coated with anti-integrin beta 1 antibodies (Fig 5). Further studies are needed to scrutinize whether the reduced adhesion in mutant cells is a consequence of an altered surface expression of the integrin or of less specific pathways, such as a reduced interaction between the membrane and the underlying actin cortex.

Finally, a notable finding in our study was that the cortex of the primary cell line (NHEK2) was significantly stiffer than the one of the immortalized WT cell line (NEB1), which agrees with the findings obtained with AFM for cell interior [21]. That indicates that the immortalization process *in vitro* causes changes that induce cell mechanical softening, which is an important feature given that many cancerous mutations undergo similar changes [33, 34]. Again, a mechanistic description of this process remains obscure.

In summary, our study showed that the cellular cortical stiffness can be examined by indenting the cells laterally with AOD-steered optical tweezers without employing any moving mechanical component, such as a cantilever or a piezo microscope stage. The method was applied to compare three keratinocyte lines. The results showed the cortical stiffness of primary keratinocytes was significantly higher than that of immortalized cells. The mutant cells were also found to be slightly stiffer than wild type cells, but this difference was less prominent than the softening reported in previous AFM experiments that probed the bulk cellular stiffness. While the study largely supports the current understanding of the role of intermediate filaments in cortical mechanics, it also indicates the need for further studies of the interactions between keratin and the cellular cortex.

## Supporting information

**S1 Fig. Validation of the experimental protocol with Cytochalasin D.** The relative stiffness of breast cancer MDA cells before and after the treatment with Cytochalasin D, which disrupts the actin skeleton and softens the cells. After the treatment, the median stiffness decreased for 25% from 156 pN/μm to 117 pN/μm (p = 0.02). The stiffness was measured at the deformation rate of 1 μm/s. The median value and the quartiles are indicated and the number of measured cells is denoted above the data points.
(TIF)

**S2 Fig. A small number of KEB7 cells exhibited some keratin filaments in the perinuclear region.** The scale bar represents 10 μm.
(TIF)

**S3 Fig. The top and the side views of a typical cell with labeled actin filaments (red) and keratin 14 (green) imaged by a confocal microscope.** (A) A typical NEB1 cell and (B) a typical KEB7 cell. In both cell types, the keratin signal is located in the cell interior underneath the cortex and there is no visible colocalization of the keratin and actin signals. The white dashed lines represent the section planes at the point of indentation experiments and scale bars represent 10 μm.
(TIF)

**S4 Fig. Western blot.** Keratin 14 protein expression in NEB1, KEB7 and NHEK2 keratinocytes were analyzed by Western blot. Quantification showed that mutant KEB7 cells exhibited approximately 5 times less keratin 14 than NEB1 cells and the latter had approximately 20% less keratin 14 than NHEK2 cells. M—MagicMark XP standard.
(TIF)

**S1 Raw image.**
(TIF)

**S1 Table. A summary of the results presented in Fig 4.**
(DOCX)

**S1 Movie. Brightfield microscopy recording of an indentation experiment.** A microbead is trapped in an optical trap, pushed into the cell with a constant velocity and then retracted. The movie spans over 2 minutes. Continuous cell remodeling during the indentation experiment is clearly visible. A thin membrane tether, extracted from the cell upon retraction, is barely noticeable.
(AVI)

**S2 Movie. Time-lapse recording of a typical NEB1 cell over the period of 3h.** There is no notable cell migration.
(AVI)

**S3 Movie. Time-lapse recording of a typical NEB1 cell over the period of 3h.** There is no notable cell migration.
(AVI)

**S4 Movie. Time-lapse recording of a typical NEB1 cell over the period of 3h.** There is no notable cell migration.
(AVI)

## Author Contributions

**Conceptualization:** Špela Zemljič Jokhadar, Mirjana Liovic, Jure Derganc.

**Data curation:** Jure Derganc.

**Formal analysis:** Špela Zemljič Jokhadar, Biljana Stojković, Mirjana Liovic, Marcos Gouveia.

**Funding acquisition:** Mirjana Liovic, Rui D. M. Travasso, Jure Derganc.

**Investigation:** Špela Zemljič Jokhadar, Biljana Stojković, Marko Vidak, Tjaša Sorčan, Marcos Gouveia.

**Methodology:** Biljana Stojković, Jure Derganc.

**Writing – original draft:** Špela Zemljič Jokhadar, Mirjana Liovic, Rui D. M. Travasso, Jure Derganc.

**Writing – review & editing:** Jure Derganc.

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
