## [Decision Letter · Decision Letter 0]

1 Jun 2020

PONE-D-20-08552

Cortical stiffness of keratinocytes measured by lateral indentation with optical tweezers

PLOS ONE

Dear Dr. Zemljič Jokhadar,

Thank you for submitting your manuscript to PLOS ONE. After careful consideration, we feel that it has merit but does not fully meet PLOS ONE’s publication criteria as it currently stands. Therefore, we invite you to submit a revised version of the manuscript that addresses the points raised during the review process.

Two experts have reviewed the manuscript and found that the current study needs to be more clarified with more controls and qualified images enough to be published in the journal.

We look forward to receiving your revised manuscript.

Kind regards,

Jung Weon Lee, Ph.D.

Academic Editor

PLOS ONE

Journal Requirements:

2. Please provide additional information about the NEB1, KEB7, NHEK2 and MDA-MB-231 cell lines used in this work, including the source of the cell lines and any quality control testing procedures (authentication, characterisation, and mycoplasma testing). For more information, please see http://journals.plos.org/plosone/s/submission-guidelines#loc-cell-lines.

Reviewers' comments:

Reviewer's Responses to Questions

**Comments to the Author**

1. Is the manuscript technically sound, and do the data support the conclusions?

Reviewer #1: Partly

Reviewer #2: Partly

2. Has the statistical analysis been performed appropriately and rigorously? 

Reviewer #1: Yes

Reviewer #2: Yes

3. Have the authors made all data underlying the findings in their manuscript fully available?

Reviewer #1: Yes

Reviewer #2: Yes

4. Is the manuscript presented in an intelligible fashion and written in standard English?

Reviewer #1: Yes

Reviewer #2: Yes

5. Review Comments to the Author

Reviewer #1: The cytokeratin pattern of human skin keratinocytes is well known. Despite a series of publications the exact influence of cytokeratins on cellular mechanical properties needs further elucidation to understand their functional role. The authors isolated primary cells from healthy persons (NHEK2 ), immortalized wild type cells (NEB1) and isolated and immortalized cells (KEB7) with K14 R125P mutant. Using optical tweezers a small glass ball is moved towards one side of these cells at two different speeds. The shift of the ball out of the center of the beam gives a measure of the force which is applied to the cell by the touching ball. This procedure is performed once or twice at the same position. The stiffness as determined by this indentation differs between the three cell types, being highest in the primary cells and lowest in the immortalized cells, slightly higher in the mutant cells. The values of the immortalised cells vary significantly less than those of the other cells, including fixed cells which act as controls. The slow moving bead is facing higher stiffness than when approached fast (0,1 versus 1 µm/s) The measurements are not sensitive to cytochalasin.

The biomechanical measurements are convincing and elegant. Whether the cells are observed using bright field microscopy and not phase contrast or DIC remains an open question. Better optical contrast would have allowed to follow cell behaviour clearly and the size of the bead will be hidden by refraction anyway. The main shortcomings of this manuscript is the lack of functional aspects, on the role of the various cytoskeletal elements for these mechanical properties. At least some good fluorescence images of F-actin, cytokeratin and microtubules would be required to see what is going on on the sides where the ball is touching. This area strongly depends on adhesivity of the cells and the speed of cell movement. The cortical actin layer may be almost missing and F-actin may appear in form of ordered bundles (fast movement and strong adhesion at rear end), or in very slowly moving cells a well developed cortical net of fine fibrillar actin may be apparent. How close are the cytokeratins in these areas, are the different cell types moving at different speed and do they keep direction? In case the cells would be turning around the cortical stiffness strongly depends on the direction of the turn as was shown by acoustic microscopy of keratinocytes.

Reviewer #2: Most of eukaryotic cells lack a cell wall and the main determinant of the cell surface stiffness is represented by cellular cortex which is a thin crosslinked actin network that lies immediatly under the plasma membrane. In the last years the cortex has received increasing attention due to its functions in many cellular processes including cytokinesis and cell migration.

Local mechanical probing has shown that actin rules cortical stifness whereas intermediate filaments, namely vimentin, contribute mainly to cytoplasmic stifness.

In the manuscript by Zemljic s. et al. the authors attempted to assess the role played by keratin filaments in regulating the keratinocytes cortical stifness by employing a unusual approach involving the use of lateral indentation with optical tweezers. They compared the cortical stifness of three different keratinocyte cell lines alongside to their aptitude to tune focal adhesions by quantifying the bead adhesion events in indentation experiments.

Prior publications there are some major and minor demanding issues that have to be sorted-out.

Major

1) The three cell lines displayed different cortical stiffness. However, as the authors mentioned, the cell lines are quite different being 2 out of three immortalized and one primary keratinocytes (NHEK2). Could you please provide a Western Blot showing the amount of Keratin 14 for each cell line when compared to a suitable control?

2) The authors state that keratins have been shown to partecipate in focal adhesion thus they assess ahether the mutated keratin affect the process they quantified the adhesion events in indentation experiments by using beads coated with anti-integrin B1 antibody. Would it be possible to coat the beads with an extracellular matrix component (e.g. collagen) which represents the physiological ligand for integrin?

3) Table S1 and fig.3: while for NEB1 cells measurements are quite clusterd for the others they are quite spread. Could you please implement data for NHEK2 cells at the condition of 1um/s and those of fixed NEB1with at least 35 measurements? Furthermore, would it be possible implement the experiment by increasing twice the indentation deep (400 nm instead of 200 nm)?

4) I understand that the crosstalk between inetermediate- and actin-filaments is mostly unknown. However, it would be greatly appreciated whether the authors could discuss a little bit this aspect in the discussion.

Minor

1)Concerning the material and methods section the author should provide details regarding the medium used to culture cells. a) Does the EpiLife medium contain, or not, calcium? if yes how much? b) How much is diluted the EpiLife defined growth supplement? Is plasticware coated? if yes what with has been coated?

2)Few typos.

3)When it comes to the section "Results" (pag. 7 line 172) could you please clarify what is the meaning of k1 and k2.

4)It may be that I missed that point but for how long and what is the concentration that has been used for CytochalisnD experiment?

6. PLOS authors have the option to publish the peer review history of their article (what does this mean?). If published, this will include your full peer review and any attached files.

Reviewer #1: Yes: Juergen Bereiter-Hahn, Institute for Neurosciences and Cell Biology, Goethe University Frankfurt

Reviewer #2: No

---

## [Author Response · Author response to Decision Letter 0]

18 Sep 2020

Reviewer #1: The cytokeratin pattern of human skin keratinocytes is well known. Despite a series of publications the exact influence of cytokeratins on cellular mechanical properties needs further elucidation to understand their functional role. The authors isolated primary cells from healthy persons (NHEK2 ), immortalized wild type cells (NEB1) and isolated and immortalized cells (KEB7) with K14 R125P mutant. Using optical tweezers a small glass ball is moved towards one side of these cells at two different speeds. The shift of the ball out of the center of the beam gives a measure of the force which is applied to the cell by the touching ball. This procedure is performed once or twice at the same position. The stiffness as determined by this indentation differs between the three cell types, being highest in the primary cells and lowest in the immortalized cells, slightly higher in the mutant cells. The values of the immortalised cells vary significantly less than those of the other cells, including fixed cells which act as controls. The slow moving bead is facing higher stiffness than when approached fast (0,1 versus 1 µm/s) The measurements are not sensitive to cytochalasin.

1. The biomechanical measurements are convincing and elegant. Whether the cells are observed using bright field microscopy and not phase contrast or DIC remains an open question. Better optical contrast would have allowed to follow cell behaviour clearly and the size of the bead will be hidden by refraction anyway. 

We thank the reviewer for the positive comments. Our current optical tweezers set-up only supports normal bright field imaging. Fortunately, even with this technique the contrast was good enough so that most of the time we were able to perform the indentation experiments by avoiding lamellipodia that would interfere with the measurement. For our future studies, we would like to upgrade the microscope with an external phase contrast module, which would be compatible with the laser and would thus expand the usability of the set-up. The missing information about the employed microscopy technique has been now added to Materials and methods and in the Figure caption.

2. The main shortcomings of this manuscript is the lack of functional aspects, on the role of the various cytoskeletal elements for these mechanical properties. At least some good fluorescence images of F-actin, cytokeratin and microtubules would be required to see what is going on on the sides where the ball is touching. This area strongly depends on adhesivity of the cells and the speed of cell movement. The cortical actin layer may be almost missing and F-actin may appear in form of ordered bundles (fast movement and strong adhesion at rear end), or in very slowly moving cells a well developed cortical net of fine fibrillar actin may be apparent. How close are the cytokeratins in these areas, are the different cell types moving at different speed and do they keep direction? In case the cells would be turning around the cortical stiffness strongly depends on the direction of the turn as was shown by acoustic microscopy of keratinocytes.

We thank the reviewer for the observation that the information on the state of all cytoskeletal elements is crucial for interpretation of mechanical properties of cells. While a detailed scrutiny of the interplay between different cytoskeletal elements is beyond the scope of this paper (and, in fact, our experimental capacity), we performed confocal imaging to visually examine the integrity of the cytoskeletal elements in our cells. These new results are now presented in a new section »Distribution of cytoskeletal elements« with additional figures (Fig 2 and Fig S2), and a description of immunofluorescence is added to Materials and methods. The results showed that all the cell lines exhibited the expected behavior, i.e., the keratin filaments were disrupted only in the mutant keratin cell line KEB7. Also, within the resolution of our confocal microscope, actin filaments and microtubules exhibited normal morphology and the sub-membrane actin was present at the point of the bead indentation in all three cell lines. 

We thank the reviewer for pointing out the relation between cell migration and their mechanical properties, which we did not explicitly consider in our first submission. As noted by the reviewer, the leading edge of migrating keratinocytes markedly differs from their trailing edge in terms of cell stiffness (Bereiter-Hahn & Lüers, 1998) and membrane tension (Lieber et al., 2015). In our indentation experiments, which were performed within approximately 30 min, we did not observe notable cell migration with any cell line. In fact, very few cells exhibited the typical morphology of migrating cells, i.e., the one resembling a fried egg cut into a crescent shape with distinct leading/trailing edges. To examine cell behavior over longer periods of time we have now also performed time-lapse recording of cells. These experiments confirmed that, unlike the frog and fish keratinocytes that were used in the above studies, our cells did not exhibit notable movements and rather extended lamellipodia in all directions. These points are now emphasized in Discussion together with additional references. To avoid misunderstanding, we changed the terms leading/trailing edge to a more descriptive ones (e.g., “trailing edge” -> “cell side without lamellipodia”). Time-lapse recordings of typical cells have been added to the Supporting information. Also, we now clearly state that out mechanical experiments were performed on individual non-confluent cells.

Following the suggestions made by the reviewer, we further investigated a possible connection between the mutated keratin and the cortical stiffness. Namely, previous studies on live GFP-transfected mutant keratinocytes found highly dynamic keratin particles in the cell periphery, that could in principle interact with the cortex [refs 7-10]. We therefore performed additional confocal microscopy to compare distributions of actin and keratin in NEB1 and KEB7 cells. The results are presented in the Supporting material (Fig S3). In both cell types, the keratin signal was distributed in the cell interior underneath the actin cortex. Within our optical resolution, no larger keratin particles and no clear interactions between the keratin and the cortex were detected. These aspects are now further addressed in the Discussion (see also the answer to the comment #4 made by reviewer #2).

Additional references that were added, regarding the mechanics of migrating keratinocytes:

Bereiter-Hahn J, Lüers H. Subcellular tension fields and mechanical resistance of the lamella front related to the direction of locomotion. Cell Biochem Biophys. 1998;29(3):243-62. Epub 1998/12/30. doi: 10.1007/bf02737897. PubMed PMID: 9868581.

Lieber AD, Schweitzer Y, Kozlov MM, Keren K. Front-to-rear membrane tension gradient in rapidly moving cells. Biophys J. 2015;108(7):1599-603. Epub 2015/04/12. doi: 10.1016/j.bpj.2015.02.007. PubMed PMID: 25863051; PubMed Central PMCID: PMCPMC4390806.

Reviewer #2: Most of eukaryotic cells lack a cell wall and the main determinant of the cell surface stiffness is represented by cellular cortex which is a thin crosslinked actin network that lies immediatly under the plasma membrane. In the last years the cortex has received increasing attention due to its functions in many cellular processes including cytokinesis and cell migration.

Local mechanical probing has shown that actin rules cortical stifness whereas intermediate filaments, namely vimentin, contribute mainly to cytoplasmic stifness.

In the manuscript by Zemljic s. et al. the authors attempted to assess the role played by keratin filaments in regulating the keratinocytes cortical stifness by employing a unusual approach involving the use of lateral indentation with optical tweezers. They compared the cortical stifness of three different keratinocyte cell lines alongside to their aptitude to tune focal adhesions by quantifying the bead adhesion events in indentation experiments.

Prior publications there are some major and minor demanding issues that have to be sorted-out.

Major

1. The three cell lines displayed different cortical stiffness. However, as the authors mentioned, the cell lines are quite different being 2 out of three immortalized and one primary keratinocytes (NHEK2). Could you please provide a Western Blot showing the amount of Keratin 14 for each cell line when compared to a suitable control?

We thank the reviewer for the constructive remarks regarding our study and we acknowledge that western blot of keratins is important information that is needed to interpret the results of mechanical experiments. We therefore performed western blot showing keratin K14 in all three utilized cell lines (in our study, NEB1 cells served as control cells as the main focus was to measure if the impaired keratin affects the cortical stiffness). The results showed that NEB1 and NHEK2 cells had much more keratin 14 than the mutant KEB7 cell (Fig S4). We quantified these results as NHEK2 >NEB1>KEB7, where there is about 30% less keratin in NEB1 cells than in NHEK2, and about 10x less keratin in KEB7 cells than in NEB1. The western blot has been now added to the Supporting information (Fig S4). 

These new results support the implicit assumption that was made in our first submission, i.e., that the major difference between cell lines is the severely impaired keratin in mutant cells. These new data is in line with the main result of our study, namely that keratin is not a major player in cortical stiffness. 

2. The authors state that keratins have been shown to partecipate in focal adhesion thus they assess ahether the mutated keratin affect the process they quantified the adhesion events in indentation experiments by using beads coated with anti-integrin B1 antibody. Would it be possible to coat the beads with an extracellular matrix component (e.g. collagen) which represents the physiological ligand for integrin?

We thank the reviewer for this recommendation. In our study we anticipated that the side-indentation of cells with optical tweezers could be useful for assessing cell-bead adhesion. We therefore employed the anti-integrin B1 antibody because we could covalently attached it to our streptavidin-coated silica beads via a biotin-streptavidin bond. Following the reviewer’s suggestion we explored also the possibility to coat our beads with collagen or fibronectin via non-specific binding, but in the given time-frame we could not develop a robust experimental protocol that would ensure reliable experimental results (e.g., we were not able to develop a reliable control experiment for assessing the extent of coating). 

Encouraged by the reviewer’s comment, we are now even more convinced that our protocol would be particular suitable for adhesion experiments because the side-indentation into a cell is gentle and can be well controlled. We are therefore developing an upgraded tweezers set-up with real-time force measurement and feed-back that would allow a better control over cell-bead contact times (see also our response to the next question) and look forward to implementing also alternative bead coatings. 

3. Table S1 and fig.3: while for NEB1 cells measurements are quite clusterd for the others they are quite spread. Could you please implement data for NHEK2 cells at the condition of 1um/s and those of fixed NEB1 with at least 35 measurements? Furthermore, would it be possible implement the experiment by increasing twice the indentation deep (400 nm instead of 200 nm)?

We acknowledge that there is a seemingly puzzling discrepancy in the number of the measured cells for each cell type. The reason for it lies in our experimental protocol, which proved to be quite elaborate and time consuming. Namely, in the presented experimental set-up, the force on the bead was not measured in real time but only after the indentation was completed. We therefore could not implement a feed-back mechanism that would automatically retract the bead when the force (or indentation depth) reached a certain value. The contact point between the bead and the cell was adjusted manually, based on real-time visual inspection (in the experiment, the bead was moving to-and-fro a predefined point with a constant speed and the cell was slowly moved towards the same predefined point using a manual the microscope stage until the bead started touching the cell). Because we aimed to indent the cells as gently as possible (to be able to repeat two measurements on the same spot), many of the trials ended without a usable force-indentation curve. In addition, many of the indentations had to be discarded because free-floating debris, which is always present with living cells, was attracted into the laser trap and ruined the force measurement. 

To assure the same experimental conditions for all three cell lines, the experiments were carried in parallel in the same time period. We aimed to collect approximately 20 data points for each cell, which is comparable to the number of data points that are typically collected in similar studies. However, it turned out that the primary cell line was much more difficult to work with, and by the time we had collected 20 points for NHEK2 cells, we ended up with many more data points on the NEB1 and KEB7 cell lines. Fortunately, this, in turn, vastly improved the statistics for these two cell types, which were in the focus of our study.

We have now amended the manuscript to emphasize the above-mentioned drawbacks of our experimental protocol. Also, we are now developing an enhanced protocol with a real-time force measurement that would greatly improve our experimental throughput and provide a much better control over the maximum indentation depth (or maximum indentation force). Such a protocol would also allow to reliably acquire and analyze the force-indentation curves at greater indentation depths, as suggested by the reviewer. 

4. I understand that the crosstalk between inetermediate- and actin-filaments is mostly unknown. However, it would be greatly appreciated whether the authors could discuss a little bit this aspect in the discussion.

We agree with both reviewers (see also the comment #2 by reviewer #1) that the ultimate goal of studies like ours should be a better understanding of the functional role of keratins in cell mechanics. As described in our answer to the comment #2 by reviewer #1, we performed additional imaging to explore possible connections between actin and keratin. These aspects are now further addressed in the Discussion. A reference to a recent review paper on the cytoskeletal cross-talk has been added (Seetharaman et al. 2020).

Seetharaman, Shailaja, and Sandrine Etienne-Manneville. "Cytoskeletal crosstalk in cell migration." Trends in Cell Biology (2020).

Minor

1. Concerning the material and methods section the author should provide details regarding the medium used to culture cells. a) Does the EpiLife medium contain, or not, calcium? if yes how much? b) How much is diluted the EpiLife defined growth supplement? Is plasticware coated? if yes what with has been coated?

The missing data was added to the manuscript.

2. Few typos.

The manuscript was proofread by a professional English speaker. 

3. When it comes to the section "Results" (pag. 7 line 172) could you please clarify what is the meaning of k1 and k2.

The missing data was added to the manuscript.

4. It may be that I missed that point but for how long and what is the concentration that has been used for CytochalisnD experiment?

The missing data was added to the manuscript.

---

## [Decision Letter · Decision Letter 1]

6 Oct 2020

PONE-D-20-08552R1

Cortical stiffness of keratinocytes measured by lateral indentation with optical tweezers

PLOS ONE

Dear Dr. Zemljič Jokhadar,

Thank you for submitting your manuscript to PLOS ONE. After careful consideration, we feel that it has merit but does not fully meet PLOS ONE’s publication criteria as it currently stands. Therefore, we invite you to submit a revised version of the manuscript that addresses the points raised during the review process.

One of the previous reviewers found that certain minor points should clarified before a publication in the journal including typos.

We look forward to receiving your revised manuscript.

Kind regards,

Jung Weon Lee, Ph.D.

Academic Editor

PLOS ONE

Reviewers' comments:

Reviewer's Responses to Questions

**Comments to the Author**

1. If the authors have adequately addressed your comments raised in a previous round of review and you feel that this manuscript is now acceptable for publication, you may indicate that here to bypass the “Comments to the Author” section, enter your conflict of interest statement in the “Confidential to Editor” section, and submit your "Accept" recommendation.

Reviewer #1: All comments have been addressed

Reviewer #2: (No Response)

2. Is the manuscript technically sound, and do the data support the conclusions?

Reviewer #1: Yes

Reviewer #2: Partly

3. Has the statistical analysis been performed appropriately and rigorously? 

Reviewer #1: Yes

Reviewer #2: Yes

4. Have the authors made all data underlying the findings in their manuscript fully available?

Reviewer #1: (No Response)

Reviewer #2: Yes

5. Is the manuscript presented in an intelligible fashion and written in standard English?

Reviewer #1: Yes

Reviewer #2: Yes

6. Review Comments to the Author

Reviewer #1: Fig 2 shows a KEB7 cell with very sparse F-actin, this does not support the statement line 196/197 that the cytoskeleton of these cells differs only in the lack of keratin 14 expression. However in S3 a KEB7 cell is shown that expresses well developed F-actin. I propose to mention this variability in the legend to Fig. 2.

The final question of the role of cytokeratin in cell mechanics has not been solved and will require exensive further work. The very high variability of the reaction data would support the hypothesis that cytokeratin stabilises F-actin arrangements and by this supports more homogenous reactions. But anyway, this is a very crude hypothesis which also needs intensive research. Therefore we can accept that the paper does not try to extend our knowledge by extensive hypotheses.

Reviewer #2: I truly appreciated the efforts made by Jokhadar and colleagues and I understand that they encountered some kind of technical difficulties in setting up robust protocol, due to the time window, to properly address the arguments raised by the reviewer. On the whole the manuscript has been improved. However, in my opinion, before publication a couple of minor issues should be settled.

1) When comparing protein abundance across different samples (e.g. different cell lines), it is critical to have a method to account for variation due to errors in loading or later during the transfer. Thus, I am wondering how can be possible that the authors could estimate the amount of a given protein in a Western Blot without of a proper loading control? Please provide a proper loading control (e.g. GAPDH, actin, tubulin, lamin, vinculin etc…). Only after having assessed the band intensities of both proteins (housekeeping and keratin 14) one can draw conclusions and compare the protein amount among different samples.

2) Please check once more the manuscript because throughout the main text there are still few typos that should be edited (e.g. 12.5 rpm -> either 12.5 Krpm or 12,500 rpm)

3) In the section material and methods it would be nice to indicate the catalogue number in addition to the company from which the authors purchased the antibodies used in the study.

7. PLOS authors have the option to publish the peer review history of their article (what does this mean?). If published, this will include your full peer review and any attached files.

Reviewer #1: **Yes: **Juergen Bereiter-Hahn

Reviewer #2: No

---

## [Author Response · Author response to Decision Letter 1]

10 Nov 2020

Reviewer #1

 Fig 2 shows a KEB7 cell with very sparse F-actin, this does not support the statement line 196/197 that the cytoskeleton of these cells differs only in the lack of keratin 14 expression. However in S3 a KEB7 cell is shown that expresses well developed F-actin. I propose to mention this variability in the legend to Fig. 2.

The final question of the role of cytokeratin in cell mechanics has not been solved and will require exensive further work. The very high variability of the reaction data would support the hypothesis that cytokeratin stabilises F-actin arrangements and by this supports more homogenous reactions. But anyway, this is a very crude hypothesis which also needs intensive research. Therefore we can accept that the paper does not try to extend our knowledge by extensive hypotheses.

We thank the reviewer for pointing out this discrepancy. After reviewing the data, we realized that the weak intensity of actin signal in KEB7 cells in Fig. 2 was a result of an error in transferring the raw images into a single panel. We therefore re-edited Fig. 2 and double-checked that all the images appear with their original intensity. 

Nevertheless, we mentioned in the results (line 200) that the distribution of the cytoskeletal elements differs moderately even among cells of the same cell type. We also added to the legend of Fig 2 that the images were taken at the height were the bead indents the cell, which is approximately 5 µm above the bottom of the chamber.

Reviewer #2 

1) When comparing protein abundance across different samples (e.g. different cell lines), it is critical to have a method to account for variation due to errors in loading or later during the transfer. Thus, I am wondering how can be possible that the authors could estimate the amount of a given protein in a Western Blot without of a proper loading control? Please provide a proper loading control (e.g. GAPDH, actin, tubulin, lamin, vinculin etc…). Only after having assessed the band intensities of both proteins (housekeeping and keratin 14) one can draw conclusions and compare the protein amount among different samples.

We thank the reviewer for this constructive remark and we repeated the WB once more with GAPDH as the control. As we had three independent experiments now, the results present the mean over all of them.

2) Please check once more the manuscript because throughout the main text there are still few typos that should be edited (e.g. 12.5 rpm -> either 12.5 Krpm or 12,500 rpm)

We checked the manuscript for the typos (the mentioned numbers were corrected to 12,500 rpm).

3) In the section material and methods it would be nice to indicate the catalogue number in addition to the company from which the authors purchased the antibodies used in the study.

We added the catalogue numbers of the antibodies used.

---

## [Decision Letter · Decision Letter 2]

25 Nov 2020

PONE-D-20-08552R2

Cortical stiffness of keratinocytes measured by lateral indentation with optical tweezers

PLOS ONE

Dear Dr. Zemljič Jokhadar,

Thank you for submitting your manuscript to PLOS ONE. After careful consideration, we feel that it has merit but does not fully meet PLOS ONE’s publication criteria as it currently stands. Therefore, we invite you to submit a revised version of the manuscript that addresses the points raised during the review process.

One of the previous reviewers still commented that certain points in the revised manuscript should be clarified for a publication in the journal.

We look forward to receiving your revised manuscript.

Kind regards,

Jung Weon Lee, Ph.D.

Academic Editor

PLOS ONE

Reviewers' comments:

Reviewer's Responses to Questions

**Comments to the Author**

1. If the authors have adequately addressed your comments raised in a previous round of review and you feel that this manuscript is now acceptable for publication, you may indicate that here to bypass the “Comments to the Author” section, enter your conflict of interest statement in the “Confidential to Editor” section, and submit your "Accept" recommendation.

Reviewer #2: All comments have been addressed

2. Is the manuscript technically sound, and do the data support the conclusions?

Reviewer #2: Yes

3. Has the statistical analysis been performed appropriately and rigorously? 

Reviewer #2: Yes

4. Have the authors made all data underlying the findings in their manuscript fully available?

Reviewer #2: Yes

5. Is the manuscript presented in an intelligible fashion and written in standard English?

Reviewer #2: Yes

6. Review Comments to the Author

Reviewer #2: I acknowledge the efforts made by the authors. Indeed, the authors fulfilled the reviewer’s requirements. However, I still noticed that there are a couple of minor concerns, looking like kind of discrepancies, that are deserving some attention. Examples are given below:

1) Lines 218-224: the figures provided by the authors about Western Blot keratin 14 quantification differ from those that appear in the Supporting Information legend to figure S4 (lines 544-547). Which one should I trust?

2) In the section material & methods when it comes the “Protein extraction and Western blotting” line 117 it is not clear whether the SDS-PAGE is a gradient gel (4-12%) or it is a 20% SDS-PAGE.

Please clarify these minor issues.

7. PLOS authors have the option to publish the peer review history of their article (what does this mean?). If published, this will include your full peer review and any attached files.

Reviewer #2: No

---

## [Author Response · Author response to Decision Letter 2]

1 Dec 2020

We would like to thank the Reviewer #2 for his very thorough overview of our manuscript. Both mistakes he pointed out were the results of our inaccuracy.

• Lines 218-224: the values written in these lines are correct and represent the mean of three independent Western Blots we made. In the lines 544-547 we forgot to update the numbers, which we corrected now.

• Line 117: SDS-PAGE is a gradient gel (4-12%) and not a 20% SDS-PAGE, which we corrected now.

---

## [Decision Letter · Decision Letter 3]

7 Dec 2020

Cortical stiffness of keratinocytes measured by lateral indentation with optical tweezers

PONE-D-20-08552R3

Dear Dr. Zemljič Jokhadar,

We’re pleased to inform you that your manuscript has been judged scientifically suitable for publication and will be formally accepted for publication once it meets all outstanding technical requirements.

Kind regards,

Jung Weon Lee, Ph.D.

Academic Editor

PLOS ONE

Additional Editor Comments (optional):

Reviewers' comments:

Reviewer's Responses to Questions

**Comments to the Author**

1. If the authors have adequately addressed your comments raised in a previous round of review and you feel that this manuscript is now acceptable for publication, you may indicate that here to bypass the “Comments to the Author” section, enter your conflict of interest statement in the “Confidential to Editor” section, and submit your "Accept" recommendation.

Reviewer #2: All comments have been addressed

2. Is the manuscript technically sound, and do the data support the conclusions?

Reviewer #2: Yes

3. Has the statistical analysis been performed appropriately and rigorously? 

Reviewer #2: Yes

4. Have the authors made all data underlying the findings in their manuscript fully available?

Reviewer #2: Yes

5. Is the manuscript presented in an intelligible fashion and written in standard English?

Reviewer #2: Yes

6. Review Comments to the Author

Reviewer #2: I am very grateful to the authors for their endless efforts aimed to improve their manuscript. The minor issues that deserved attention have now been properly addressed. Hence, their manuscript is now worthy of publication in the PLOS ONE Journal.

7. PLOS authors have the option to publish the peer review history of their article (what does this mean?). If published, this will include your full peer review and any attached files.

Reviewer #2: No

---

## [Editor Report · Acceptance letter]

21 Dec 2020

PONE-D-20-08552R3 

Cortical stiffness of keratinocytes measured by lateral indentation with optical tweezers 

Dear Dr. Derganc:

I'm pleased to inform you that your manuscript has been deemed suitable for publication in PLOS ONE. Congratulations! Your manuscript is now with our production department. 

Kind regards, 

on behalf of

Dr. Jung Weon Lee 

Academic Editor

PLOS ONE